# A Multifactorial Approach to Untangle Graphene Oxide (GO) Nanosheets Effects on Plants: Plant Growth-Promoting Bacteria Inoculation, Bacterial Survival, and Drought

**DOI:** 10.3390/nano11030771

**Published:** 2021-03-18

**Authors:** Tiago Lopes, Catarina Cruz, Paulo Cardoso, Ricardo Pinto, Paula A. A. P. Marques, Etelvina Figueira

**Affiliations:** 1Department of Biology, University of Aveiro, 3810-193 Aveiro, Portugal; tslopes@ua.pt (T.L.); catarinasilvacruz@ua.pt (C.C.); 2Centre for Environmental and Marine Studies, Department of Biology & CESAM, University of Aveiro, 3810-193 Aveiro, Portugal; pjcardoso@ua.pt (P.C.); rl.pinto@ua.pt (R.P.); 3Centre for Mechanical Technology and Automation, Department of Mechanics & TEMA, University of Aveiro, 3810-193 Aveiro, Portugal; paulam@ua.pt

**Keywords:** biostimulants, plant growth-promoting bacteria (PGPB), graphene oxide (GO) nanosheets, maize, drought

## Abstract

Drought is a limiting factor for agricultural productivity. Climate change threatens to expand the areas of the globe subjected to drought, as well as to increase the severity and duration of water shortage. Plant growth-promoting bacteria (PGPB) are widely studied and applied as biostimulants to increase plant production and to enhance tolerance to abiotic and biotic constraints. Besides PGPB, studies on the potential of nanoparticles to be used as biostimulants are also thriving. However, many studies report toxicity of tested nanoparticles in bacteria and plants in laboratory conditions, but few studies have reported effects of nanoparticles towards bacterial cells and communities in the soil. The combined application of nanoparticles and PGPB as biostimulant formulations are poorly explored and it is important to unravel the potentialities of their combined application as a way to potentiate food production. In this study, *Rhizobium* sp. E20-8 and graphene oxide (GO) nanosheets were applied on container-grown maize seedlings in watered and drought conditions. Bacterial survival, seedling growth (dry weight), and biochemical endpoints (photosynthetic pigments, soluble and insoluble carbohydrates, proline, lipid peroxidation, protein, electron transport system, and superoxide dismutase) were evaluated. Results showed that the simultaneous exposure to GO and *Rhizobium* sp. E20-8 was able to alleviate the stress induced by drought on maize seedlings through osmotic and antioxidant protection by GO and mitigation of GO effects on the plant’s biochemistry by *Rhizobium* sp. E20-8. These results constitute a new lead on the development of biostimulant formulations to improve plant performance and increase food production in water-limited conditions.

## 1. Introduction

The global population is projected to increase exponentially in the next decades, and to feed a growing population, food production has to keep pace with it. The expected negative impact of climate change and environmental degradation on food production will further challenge this issue. It has thus created a colossal pressure to develop new agricultural methodologies that consider food security, yield, and sustainability in a climate change scenario. Drought is a climate change factor projected to increase in the near future [1], with major impact on agricultural productivity and food security. Developing countries are particularly vulnerable to drought by lacking access to irrigation technology but also to synthetic fertilizers. A possible solution to sustainably improve crop productivity under drought stress is the application of beneficial microorganisms. Apart from having the potential to increase plant tolerance to drought [2,3], plant growth-promoting bacteria (PGPB) can help reduce fertilizer application due to their ability to improve plant growth through phosphate solubilization, nitrogen fixation, siderophore production, and phytohormone synthesis, among other traits [4,5]. For developed countries, the use of PGPB also emerges as an opportunity for a more efficient use of water and fertilizer application, reaping economic and environmental benefits. The biostimulants industry, which includes microbial inoculants, is a growing billionaire business [6]. However, available products are mostly directed to plant nutrition and plant disease biocontrol [7] and not to abiotic performance. Thus, basic research on the potential of PGPB as a way to tackle abiotic constraints, such as drought, will bring the possibility to develop new products and to widen its spectrum of use.

As food security is a matter of concern, a global effort is being made to develop new technologies for increasing food production in a sustainable way [8]. Recent developments in the field of nanotechnology have triggered increased interest in using the unique properties of nanomaterials for agriculture applications and to address major environmental challenges [9,10]. One of the materials capturing the attention of researchers is graphene oxide (GO), a two-dimensional layer of carbon atoms arranged in a hexagonal crystalline structure [11], with a high density of oxygen functional groups (carboxyl, hydroxyl, carbonyl, and epoxy) in the carbon lattice [8]. The hydrophilic nature of GO will adsorb water into the lamellar structure and allow GO to form stable suspensions in aqueous media [12,13,14]. Currently, most research is focused on the effects of GO on humans, small mammals, invertebrates, and aquatic organisms, and little research has been devoted to plants [15] or to the application of GO in agriculture [16]. Understanding the cross-talk between nanomaterials (as GO) and plants is important to manipulate the effect of this nanomaterial on agricultural contexts [16]. Research supports that GO significantly affected seed germination, root length, leaf number, shoot length, plant height, chlorophyll content, enzymatic activity, and amino acid composition [17], some of them being contradictory. However, Juárez-Maldonado et al. [18] reported that almost every nano-compound could be a biostimulant, depending only on the concentration and the physical and chemical properties in order to avoid detrimental effects on plants. Moreover, the beneficial effects of PGPB on plants may be at risk since exposure to GO was shown to have antibacterial properties. Perturbation of bacterial cell membrane by GO was demonstrated by the decrease in trans-membrane potential and the leakage of intracellular electrolytes in bacteria exposed to GO [19]. Furthermore, the oxidative stress induced by GO in bacteria was repeatedly reported [19,20,21,22,23,24,25]. These effects can decrease bacterial survival [26] and may compromise plant–microbe interactions. 

Studies on the effects of GO on plants and microorganisms are accumulating, however, many use very high and unrealistic concentrations or use hydroponic systems and in vitro cultures, which are not representative of field conditions. On the other hand, the effects of GO are determined by sole exposures, or when co-exposures are studied, they generally involve metals [27,28,29,30]. Considering that field conditions (biotic and abiotic) are constantly varying and GO effects at different abiotic conditions were not evaluated, studies evaluating GO impact in multifactorial contexts are important and will deliver information that is crucial to broaden the knowledge about GO effects before its use in agriculture can be considered.

The present study aimed to cover this gap by clarifying GO nanosheet effects on the early stage of plant growth. When the root system is starting to develop, plants are particularly vulnerable to drought, and symbiotic interactions between plants and soil microorganisms are being established. For this, maize seedlings were grown on substrate with and without GO nanosheets, inoculated or not with a bacterium strain (plant growth promoter and tolerant to drought) in the presence and absence of drought. GO nanosheet effects were evaluated by analyzing bacterial survival, plant growth (root and shoot), and plant biochemical parameters (osmolytes, antioxidants, damage, chlorophylls, metabolic, and energy) in the tested conditions. Comparison between conditions allowed us to elucidate the interference that application of GO nanosheets may have on plant–bacterial interactions and on plant tolerance to drought.

## 2. Materials and Methods

### 2.1. Graphene Oxide Nanosheets and Other Reagents

A commercial graphene oxide (GO) nanosheet water dispersion (0.4 wt % concentration) was purchased from Graphenea, San Sebastian, Spain, and used as received. According to our characterization by atomic force microscopy (AFM, VEECO Multimode, Plainview, NY, USA), the GO nanosheets are mainly multilayer with an average thickness of 1.2 μm corresponding to 2 or 3 layers and a lateral size with a wide range of dimensions from 300 nm to 5 μm. Unless specified, other reagents were purchased from Merck, Darmstadt, Germany.

### 2.2. Bacterial Strain

*Rhizobium* sp. strain E20-8 was previously isolated from root nodules of *Pisum sativum* L. [31] and identified at the genus level [32]. The partial 16S ribosomal ribonucleic acid (rRNA) gene sequence was submitted to GenBank (accession: KY491644). This strain was previously described as osmotolerant [33] and as promoting plant growth [31]. The strain was grown in tubes containing 5 mL of yeast mannitol broth (YMB) overnight at 26 °C in an orbital shaker (150 rpm).

### 2.3. Experimental Conditions

Plastic containers were filed with 200 mL washed and autoclaved sand or with a mixture of the sand and GO nanosheet solution (4 mg/mL) in a final concentration of 0.52 mg GO/g sand. Containers were pierced on the bottom to prevent soaking. In each container, 3 *Zea mays* (Dekalb DKC 6031) seeds were sown after being soaked for 2 days in aerated water. Containers with no GO nanosheets added (only sand) were also used. After seedling emergence and when plantlets were 5 cm long (14 days), half of the containers were inoculated with 3 mL of *Rhizobium* sp. strain E20-8 culture (4.0 × 10^8^ cells/mL); the other half was inoculated with 3 mL of growth medium. After inoculation, half of the containers were watered with 10 mL deionized water 2 times a week, while the other half was not watered until the end of the experiment (14 days). Plants were grown for 28 days in greenhouse conditions at 17 ± 2 °C during the day and 13 ± 2 °C during the night, at natural light with a photoperiod of 12 hours, with 3 independent replicates for each condition, with a total of 5 replicates x 2 water conditions (watered and drought) x 2 inoculation conditions (inoculated and not inoculated) x 2 GO conditions (presence and absence), resulting in containers not inoculated and without GO nanosheets (condition Ctl), containers inoculated and without GO nanosheets (condition R), containers not inoculated and with GO nanosheets (condition GO), and containers inoculated and with GO nanosheets (condition R + GO). In each condition, half of the containers were watered, and the other half were subjected to drought.

At the end of the experiment, sand was collected in order to determine the colony forming units (CFUs) and water activity (aw). Plants were also collected; roots were washed first in tap water and after in deionized water to remove substrate particles, and then were used for dry weight determination and biochemical analysis. Samples for photosynthetic pigments were immediately used; for other biochemical parameters samples were frozen (−20 °C) until used.

### 2.4. Colony Forming Units 

For CFU determination, 1 g of sand was collected and added to 9 mL of deionized water (10^−1^ dilution). Serial dilutions (10^−2^, 10^−3^, 10^−4^, 10^−5^, and 10^−6^) were performed, and 1 mL of each dilution was plated in yeast mannitol agar (YMA) medium in triplicate and incubated for 3 days at 26 °C. Colonies were counted, and results were expressed in CFU per gram of soil (CFU/g soil).

### 2.5. Water Activity

Water activity was measured according to the method described by Gee et al. (1992) [34] using a water activity meter (HP23-AW-A with HC2-AW, Rotronic AG, Bassersdorf, Switzerland).

### 2.6. Dry Weight

We used 4 to 10 washed plants for dry weight determination. Shoots and roots were separated and dried at 60 °C until constant weight was attained. Dry weights were used to calculate growth of roots and shoots of each condition relatively to watered non-inoculated and non-GO-exposed plants.

### 2.7. Photosynthetic Pigments

Fresh samples (shoots) were milled in liquid nitrogen, followed by pestle and mortar homogenization in 80% acetone (1:2 *w*/*v*), and were allowed to rest for 45 min in the dark. Extracts were centrifuged at 4000× *g* for 5 min, and pigment content was determined following the method described by Wellburn and Lichtenthaler [35]. Absorbance was measured at 663, 646, and 470 nm, and chlorophylls a and b and carotenoids were determined using the equations proposed by Wellburn and Lichtenthaler [35]. Results were expressed in micrograms per gram of dry weight (µg/g DW).

### 2.8. Soluble and Insoluble Carbohydrates

Frozen samples (shoots and roots) were milled in liquid nitrogen, followed by pestle and mortar homogenization in 2.5 mM sulfuric acid (1:2 *w*/*v*), and were incubated at 95 °C for 1 h [36]. Extracts were centrifuged at 10,000× *g* for 4 min. The supernatant was collected and used for determination of soluble carbohydrate content and the pellet for insoluble carbohydrates (starch) using the method described by Dubois et al. [37], with some modifications. To 15 µL of sample, we added 900 µL of 98% sulfuric acid and 150 µL of 5% phenol. The mixture was then incubated for 2 h at room temperature. Samples were then centrifuged at 10,000× *g* for 5 min, the supernatant was collected, and the absorbance was measured at 492 nm. Glucose standards (1–10 mg/mL) were used. Results were expressed in milligrams of glucose per gram of dry weight (mg/g DW).

### 2.9. Proline Content

Frozen samples were milled in liquid nitrogen, followed by pestle and mortar homogenization in 3% sulfosalicylic acid, and were centrifuged (1:2 *w*/*v*) at 12,000× *g* for 10 min at 4 °C. Supernatant was collected and used for determination of proline, following the method described by Bates et al. [38] with some modifications. To 250 µL of sample, we added 250 µL of acid ninhydrin and 250 µL of glacial acetic acid. After incubation for 1h at 100 °C, the reaction was stopped by placing samples in ice. Absorbance was measured at 520 nm and proline (Sigma-Aldrich, St. Louis, MO, USA) standards (0–1 mg/mL) were used. Results were expressed in milligrams of proline per gram of dry weight (μg/g DW).

### 2.10. Lipid Peroxidation

Frozen samples were milled in liquid nitrogen, followed by pestle and mortar homogenization in 20% (*v*/*v*) trichloroacetic acid (1:2 *w*/*v*), and were centrifuged at 12,000× *g* for 10 min; then, the supernatant was collected and used for quantification of thiobarbituric acid reactive substances (TBARS) according to the methodology described by Buege and Aust [39]. Absorbance was measured at 532 nm and TBARS quantification was estimated by the molar extinction coefficient for malondialdehyde (MDA) (1.56 × 105 M^−1^ cm^−1^). Results were expressed in nmol of MDA equivalents per gram of dry weight (nmol/g DW).

### 2.11. Protein Content, Electron Transport System Activity, and Superoxide Dismutase Activity 

Frozen samples were milled in liquid nitrogen, followed by pestle and mortar homogenization in sodium phosphate buffer (50 mM sodium dihydrogen phosphate monohydrate; 50 mM disodium hydrogen phosphate dihydrate; 1 mM ethylenediaminetetraacetic acid disodium salt dihydrate (EDTA); 1% (*v*/*v*) Triton X-100; 1% (*v*/*v*) polyvinylpyrrolidone (PVP); 1 mM dithiothreitol (DTT), pH 7.0) (1:2 *w*/*v*) and centrifuged at 12,000× *g* for 10 min at 4 °C. The supernatant was collected and used immediately or stored at −80 °C for determination of protein content, electron transport system (ETS), and superoxide dismutase (SOD) activity.

Protein content was determined by the method described by Robinson and Hodgen [40]. To 50 µL of sample, we added 250 µL of biuret reaction solution. Samples were then incubated in the dark for 10 min at room temperature. Absorbance was measured at 540 nm, and bovine serum albumin (BSA) (Sigma-Aldrich, St. Louis, MO, USA) was used as standard (5 to 40 mg BSA/mL). Results were expressed in mg protein per gram of dry weight (mg/g DW).

Electron transport system (ETS) activity was measured on the basis of King and Packard [41] method, using the modifications described by Owens and King [42]. To a 37.5 µL of sample, we added 107 µL of balanced salt solution (BSS) buffer (0.13 M Tris-HCl, 0.3% (*v*/*v*) Triton X-100, pH 8.5), 35.7 µL of reduced nicotinamide adenine dinucleotide phosphate NAD(P)H (1.7 mM NADH and 250 µM NADPH), and 71.4 µL of 8 mM p-iodonitrotetrazolium (INT); the reaction started with the addition of INT. The absorbance was read at 490 nm for 10 min with intervals of 25 s. The amount of formazan formed was calculated using the molar extinction coefficient of formazan (15,900 M^−1^ cm^−1^), and the results were expressed in micromole of formazan formed per min per gram of dry weight (μmol/min/g DW). 

Superoxide dismutase (SOD) activity was determined by the conversion of nitro blue tetrazolium (NBT) by the superoxide-free radicals into NBT diformazan using the methodology described by Beachamp and Fridovich (1971) [43]. To 25 µL of sample, we added 25 µL of xanthine oxidase and 250 µL of reaction buffer with NBT, which we incubated for 20 min at room temperature with orbital rotation. Absorbance was measured at 640 nm. One unit of enzymatic activity (U) represents a 50% reduction of NBT. Results were expressed in enzymatic activity (U) per gram of dry weight (U/g DW). 

### 2.12. Statistical Analysis

All parameters tested were subjected to hypothesis testing. One-way analysis of variance (ANOVA) followed by Tukey’s test were performed using SPSS version 26.0 for macOS (SPSS Inc., Chicago, USA). The null hypothesis tested was that no significant differences existed between control and test conditions. Significant differences were considered only when *p*-value ≤ 0.05 and were identified in figures with asterisks for differences between drought and watered plants in each condition (Ctl, R, GO, and R + GO) and with different letters for differences among conditions (lowercase for drought and uppercase for watered plants).

Data from growth and biochemical parameters were used to calculate a Euclidean distance similarity matrix for roots and shoots. These similarity matrices were simplified through the calculation of the distance among centroids on the basis of the conditions, and were then submitted to ordination analysis, performed by principal coordinates (PCO). Pearson correlation vectors of biochemical parameters (correlation = 0.9) were specified as supplementary variables on the PCO graph, allowing for the identification of the descriptors that most contributed to the differences observed between the conditions tested. 

## 3. Results

### 3.1. Water Activity of Substrates

The conditions tested—control (Ctl), inoculation of PGP rhizobacteria (R), addition of GO nanosheets (GO), or a combination of the two (R + GO)—did not induce significant differences in water activity, either in substrates watered or subjected to drought. On the contrary, the level of substrate moisture (watered or drought) induced significant differences in all the conditions (Ctl, R, GO, and R + GO) (Appendix A).

### 3.2. Bacterial Survival

Although the substrate used was sterilized, plants were grown in non-axenic greenhouse conditions, and therefore, at the end of the growth period, 9.2 × 10^6^ CFU per gram of substrate (Ctl) were detected (Figure 1). In watered substrate (blue bars), inoculation with plant growth-promoting bacteria (PGPB) (R) at the beginning of the plant growth increased the number of CFU (17.5 × 10^6^ CFU/g substrate). The inclusion of GO nanosheets in the substrate (GO) increased bacterial number by 27% compared to Ctl, although not significantly, and the number of CFU in the substrate with GO nanosheets and inoculated with PGPB (R + GO) was the highest recorded, and, although not significant, was 44% higher compared to R condition. 

In the presence of drought (brown bars), the number of CFU was similar (5 × 10^6^/g substrate) between conditions R and R + GO and around fivefold higher compared to conditions Ctl and GO. In all conditions tested, drought significantly decreased CFU compared to watered substrates (Figure 1).

### 3.3. Plant Growth

In watered plants (blue bars), inoculation with PGPB (R) increased root growth (5%), yet not significantly, compared to control (Ctl). On the contrary, GO nanosheets had a significant negative effect on root growth both in non-inoculated (GO) and bacteria-inoculated (R + GO) plants (Figure 2a). In plants exposed to drought (brown bars), root growth, although not significantly different among conditions, was higher in Ctl and GO than in R and R + GO conditions, evidencing that in a drought context, inoculation with bacteria (both in presence and absence of GO) reduced root growth, although not significantly. The effect of substrate moisture at each condition evidenced that in the absence of GO (Ctl and R), growth was significantly reduced in drought compared to watered plants, but in the presence of GO (GO and R + GO), the growth reduction induced by drought was not significant (Figure 2a).

In watered plants (light blue bars), inoculation with PGPB (R) negatively affected shoot growth (10%), yet not significantly, compared to control (Ctl). GO nanosheets did not affect shoot growth, neither in non-inoculated (GO) nor in bacteria-inoculated (R + GO) plants (Figure 2b). In plants exposed to drought (light green bars) shoot was similar among R, GO, and R + GO conditions and significantly lower than in Ctl condition. Drought reduced shoot growth in all the conditions tested, but the effect was only significant in Ctl, evidencing that exposure to GO nanosheets and inoculation with bacteria were not additive, and that both alleviated the detrimental effect of drought on shoots (Figure 2b).

### 3.4. Biochemical Alterations in Roots

#### 3.4.1. Soluble and Insoluble Carbohydrates

In watered plants (blue bars), inoculation (R) did not change; GO nanosheet exposure (GO) significantly increased; and inoculation in the presence of GO (R + GO) increased, although not significantly, both soluble and insoluble carbohydrates (Figure 3a,b). In plants exposed to drought (brown bars), both carbohydrates were lower in control compared to the remaining conditions, but significant differences were only observed in soluble carbohydrates. In each condition, drought had no or a slight positive effect on carbohydrate content, except for inoculated plants (R), where soluble carbohydrates were significantly higher compared to watered inoculated ones (Figure 3a,b).

#### 3.4.2. Proline 

In watered plants, proline levels increased in all conditions relative to the control, but only in GO condition a significant increase was observed (Figure 3c). In plants exposed to drought, proline levels were higher in inoculated plants (R and R + GO) relative to the remaining conditions (Ctl and GO). Drought significantly increased proline levels in Ctl, R and R + GO conditions. On the contrary, exposure to GO caused a slight and non-significant decrease of proline in plants exposed to drought compared to watered ones.

#### 3.4.3. Protein Content

In watered plants, none of the tested conditions induced significant changes in protein levels compared to control, although inoculated plants (condition R) evidenced 32 to 42% higher protein content compared to other conditions (Figure 3d). In plants exposed to drought, a significant increase in protein levels was observed in inoculated plants (R and R + GO) compared to the remaining conditions (Ctl and GO). In each condition, drought increased protein levels, significantly in R + GO condition, around 50% in Ctl and R and negligibly (11%) in GO condition.

#### 3.4.4. ETS

Neither the level of substrate moisture nor conditions significantly changed the activity of the electron transport chain, although in the presence of drought inoculated plants (R and R + GO) showed higher levels of ETS compared to the remaining conditions (Figure 3e).

#### 3.4.5. SOD

In watered plants, both inoculation and GO nanosheets increased SOD activity, although the effect of GO was higher (GO and R + GO) and significant (GO) than inoculation when compared to Ctl (Figure 3f). In plants exposed to drought, GO nanosheets also induced the highest changes in SOD activity, being significantly different from the remaining conditions. In each condition, the effect of drought on SOD activity was negligible, evidencing that conditions had higher effect on SOD activity than moisture level. 

#### 3.4.6. Lipid Peroxidation 

In watered plants, lipid peroxidation (LPO) levels were similar among conditions (Figure 3g). In plants exposed to drought, GO nanosheet conditions (GO and R + GO) significantly induced LPO levels compared to Ctl and R. In each condition, the effect of drought increased significantly LPO levels in GO conditions (GO and R + GO), but had a marginal effect on Ctl and R.

#### 3.4.7. Multivariate Analysis 

From the multivariate analysis (Figure 3h) of root biochemical changes induced by the factors studied (GO exposure, bacteria inoculation, and drought), it is possible to observe that most biochemical markers were strongly correlated with PCO1 and therefore more related to drought conditions, confirming that drought was the factor inducing most of the biochemical changes. However, the other two factors also contributed to biochemical changes in roots. Bacterial inoculation under drought (R and R + GO) strongly correlated with the negative side of PCO1 and the positive side of PCO2 axes, evidencing that under drought, bacterial inoculation triggered mechanisms (protein and ETS) to fight osmotic stress (proline). GO nanosheet exposure strongly correlated with the negative side of both PCO1 and PCO2 axes, evidencing that root cells exposed to GO nanosheets, both in watered and drought conditions, were able to trigger the antioxidant response (SOD activity) but high correlations were also obtained for LPO (damage) (Figure 3h).

### 3.5. Biochemical Alterations in Shoots 

#### 3.5.1. Soluble and Insoluble Carbohydrates

In watered plants (light blue bars), exposure to GO nanosheets (GO and R + GO) increased soluble carbohydrates, only significantly for R + GO, relatively to Ctl and R conditions (Figure 4a). In plants exposed to drought (light green bars), the trend was the opposite, with inoculated plants exposed to GO nanosheets (R + GO) having lower levels of soluble carbohydrates than the remaining conditions, which was significantly different from Ctl. Drought had different effects on soluble carbohydrate levels, increasing significantly in Ctl and R, increasing 26% in GO, and decreasing 25% in R + GO. 

In watered plants (blue light bars), insoluble carbohydrates slightly varied among conditions (Figure 4b). In plants exposed to drought, inoculation (R) increased significantly, GO nanosheet exposure had no effect, and R + GO exposure significantly decreased the levels of insoluble carbohydrates compared to control. Although differences in the concentration of insoluble carbohydrates between watered and plants exposed to drought were detected in each condition, no significant differences were observed. 

#### 3.5.2. Proline 

In watered plants, no significant differences were observed among conditions (Figure 4c). In plants exposed to drought, proline levels were significantly higher in R condition and significantly lower in GO relative to Ctl and R + GO. Drought significantly increased proline levels in Ctl and R, but in GO and R + GO, changes were small and not significant.

#### 3.5.3. Photosynthetic Pigments 

In watered plants, conditions had little influence on the content of both chlorophylls (a and b), although a non-significant decrease was noticed in inoculated (R) plants (Figure 4d,e). In plants exposed to drought, inoculation (R condition) did not change, but GO nanosheets (GO and R + GO) decreased chlorophyll content (a and b), only significantly for chlorophyll b at GO condition. No significant differences were observed between moisture levels at each condition. However, in Ctl and R, watered plants had higher chlorophyll levels than plants exposed to drought, and the opposite was observed for GO and R + GO conditions. 

In watered plants, carotenoid levels were similar among conditions (Figure 4f). In plants exposed to drought, inoculation did not change, but GO nanosheet-exposed plants (GO and R + GO) had significantly lower carotenoid levels than control. Drought significantly increased carotenoid levels in Ctl and R conditions, but little influence was observed in GO nanosheet- and R + GO nanosheet-exposed plants.

#### 3.5.4. Protein Content

In watered plants, inoculation (R) did not influence protein levels. Exposure to GO nanosheets, especially in inoculated plants (R + GO), increased protein content (Figure 4g). In plants exposed to drought, GO nanosheet exposure had no influence, but inoculation increased protein content (R and R + GO), although not significantly. Drought increased proteins significantly in R condition. In Ctl, a non-significant increase was noticed. In GO and R + GO, drought non-significantly decreased 16% and 28% protein content, respectively.

#### 3.5.5. ETS

In watered plants, exposure to GO (GO and R + GO) significantly increased ETS activity compared to Ctl and R conditions (Figure 4h). In plants exposed to drought, ETS activity was similar among conditions. Drought increased ETS activity in Ctl and R conditions although not significantly, and marginally decreased (near 6%) the activity of ETS in GO nanosheet-exposed plants (GO and R + GO).

#### 3.5.6. SOD

In watered plants, no significant differences were found among conditions, yet SOD activity was 29 and 62% higher in GO and R + GO-exposed plants compared to Ctl, respectively (Figure 4i). In plants exposed to drought, SOD activity was higher in Ctl condition, but no significant differences were observed among Ctl, R, and R + GO conditions. Sole exposure to GO nanosheets significantly decreased SOD activity relative to Ctl and R conditions, but not to R + GO. The effect of drought on SOD activity varied among conditions, being negligible for GO and R + GO, increasing non-significantly in R condition, and increasing significantly in Ctl. 

#### 3.5.7. Lipid Peroxidation 

In watered plants, LPO was not significantly different among conditions, but increases around 33% were observed in inoculated plants (R and R + GO) compared to Ctl and GO conditions (Figure 4j). In plants exposed to drought, inoculation did not change LPO levels, but exposure to GO nanosheets decreased LPO, not significantly at R + GO, but significantly at GO comparatively to Ctl and R conditions. Drought increased LPO significantly in Ctl, not significantly in R, and had little influence in GO and R + GO conditions.

#### 3.5.8. Multivariate Analysis 

From the multivariate analysis (Figure 4k) of shoot biochemical changes induced by the factors studied (GO nanosheets exposure, bacteria inoculation, and drought), it is possible to observe that most biochemical markers were strongly correlated with PCO1, corroborating that drought and non-exposure to GO were the factors inducing most of the biochemical changes in shoots (drought in Ctl and R conditions). Exposure to GO nanosheets (GO and R + GO) correlated with the negative side of PCO2 axis, evidencing that GO nanosheets had a higher influence on shoots than bacterial inoculation, since condition R + GO was closer to GO than to R, both in watered and drought plants (Figure 4k).

## 4. Discussion

Nanomaterials are emerging as a significant breakthrough to maximize crop productivity and to minimize environmental pollution [44]. However, divergent effects for nanomaterials, including GO, on plants have been reported, compromising their use. Moreover, there is a lack of information on the effects of GO on different abiotic conditions. This study intended to clarify the effect of GO nanosheets application in an agronomic perspective by evaluating alterations in the number of PGPB in soil; assessing changes in osmotic, oxidative, and biochemical status and growth of plants; recognizing GO nanosheet co-exposure with PGPB effects on plants; and identifying changes in the tolerance of plants to abiotic factors such as drought.

Bacteria number on the substrate in condition R + GO was identical (drought) or even higher (watered) compared to condition R, showing that GO nanosheets at a relatively high concentration (0.52 mg per g soil) did not harm or even promoted the survival of bacteria in the substrate. These results differ from other studies, pointing out graphene nanomaterials, including GO, as having antibacterial activity. Du et al. [45] reported that GO decreased the abundance and diversity of beneficial endophytic bacterial populations in roots of hydroponically grown rice plants. Liu et al. [46] also observed that bacterial cells exposed to GO appeared to be completely wrapped in GO sheets and concluded that cell wrapping may limit bacterial growth by isolating cells from the medium, preventing nutrient absorption, or blocking active sites on the cell surface. However, in most studies, the reported GO effects on bacteria were evaluated by growing bacteria in liquid medium, with agitation, which imposes mechanical injury on cells. These conditions diverge from those of bacteria growing in soil. In our study, the inoculation of bacteria in substrate avoids mechanical injury originated by culture medium agitation and therefore can better assess the effects of GO presence (intentional or inadvertent) on soil bacteria communities in the field, evidencing lower detrimental effects than initially expected. 

In aqueous solutions, the water directly bonds to GO, and water molecules from the first contact layer can establish additional bonds with ‘‘free’’ water molecules from the bulk solvent [47]. Thus, hydration spheres are formed around the GO nanosheets, reducing free water, and may causing osmotic stress. However, our work showed no difference in water activity of the substrate between conditions with and without GO nanosheets (Appendix A), which can be derived from the low GO/substrate ratio (0.00053 *w*/*w*). On the other hand, Liu et al. [46] reported that GO sheets adhered to bacteria surface, hampering nutrient absorption and hindering growth. If GO sheets also adhere to roots surface, especially near root hairs, where most of the water and nutrients are taken up, absorption will be affected and may cause osmotic stress and nutritional deficiencies. In fact, Zhao et al. [48] found that GO in the range of μg/L accumulated in the root hairs of *Arabidopsis thaliana* plants, and Du et al. [45] observed plasmolysis in the root cells of rice exposed to GO, showing the osmotic stress induced by exposure to GO. Our study showed that proline and soluble carbohydrate content were higher in the roots of watered GO nanosheet-exposed plants. Soluble carbohydrates and proline are two solutes adapting plants to osmotic stress conditions [49], since they are compatible solutes that decrease the osmotic potential of cells and prevent osmotic stress and cell plasmolysis, allowing cells to absorb water from media with lower water potential [50]. Therefore, results seem to show that GO nanosheets impose osmotic stress even on watered plants.

However, in plants already exposed to GO nanosheets, drought did not appear to further increase osmotic stress, since proline levels did not increase, unlike in the remaining conditions (Ctl, R and R + GO) where proline increased significantly compared to the same watered condition. Thus, it appears that pre-exposure to GO nanosheets induced biochemical changes that pre-adapted roots to the osmotic effects of drought, and cells evidenced lower adjustments when effectively exposed to drought conditions, i.e., pre-exposure to GO nanosheets seemed to induce systemic tolerance to drought in *Z*. *mays* plants. The induction of systemic drought tolerance was already reported by Cho et al. [51] in *A*. *thaliana* plants exposed to 2,3-butanediol. Our study showed that this effect may also be induced by GO nanosheets. 

Exposure to GO nanosheets did not cause changes in root cell metabolism (proteins and ETS), and the increase in oxidative damage was reduced (21%) and not significant, which was only possible due to the increase in SOD activity that was able to quash the effects of oxidative stress induced by GO nanosheets. However, simultaneous exposure to GO nanosheets and drought induced oxidative damage in membranes (LPO increase). Cheng et al. (2016) [16] did not observe significant changes in LPO at any of the concentrations tested, but the highest concentration (100 mg GO/L) significantly increased SOD activity. Hu et al. [52] showed that GO caused metabolic disturbances linked to key biological processes, such as inhibition of carbohydrate and amino acid metabolism, and increase of unsaturated to saturated fatty acids ratio. Liu et al. [53] and Liu et al. [54] indicated that GO treatment resulted in a high concentration of ROS and affected the anabolism of *A*. *thaliana* plants. Zhou and Hu [55] reported that upregulation of antioxidant enzymes (such as SOD and peroxidase) and increase of LPO content suggested that GO induced oxidative stress that caused damage in plant cells. These results are in agreement with the oxidative damage induced by GO nanosheets in the roots of *Z*. *mays* observed in our study. In the presence of GO nanosheets and drought, root cells were unable to further increase SOD activity nor to manage a higher level of stress, as LPO increase evidenced. Thus, if pre-exposure to GO seemed to adapt roots to the osmotic effects of drought, the oxidative stress was not effectively abrogated and damage was not overcome, although the impact on cell metabolism was low.

Our study evidenced that in the shoot, GO nanosheets showed no signs of causing oxidative or osmotic stress. However, influence on the general cell metabolism (proteins) and on the use of energy (ETS) was noticed, suggesting the induction of mechanisms adapting cells to the new prevailing conditions. GO nanosheets seemed to have protected shoots from the drought effects, since proline and soluble carbohydrates levels and SOD activity in the presence of drought were identical or slightly lower than at the same watered condition. However, under drought insoluble carbohydrates (storage energy), chlorophyll and especially carotenoid content decreased in the presence of GO (conditions GO and R + GO). The decrease in chlorophylls was already reported as an effect of exposure to GO [56,57,58]. The decrease in chlorophylls and carotenoids originated by GO, observed in our study, may be related to resources diversion for the synthesis of other metabolically related compounds such as ABA, which have already been reported to increase in plants exposed to GO [16]. Since ABA induces stomata closure [49], reducing gas exchange and CO_2_ concentration in the mesophyll, the Calvin cycle activity and photosynthates production will be lower [49], leading to a decrease in storage energy (insoluble carbohydrates). In fact, our results showed that insoluble carbohydrate (starch) levels were lower in plants exposed to GO nanosheets and drought. Our study also evidenced that in presence of drought, oxidative damage of shoot cells was higher in the absence of GO nanosheets, even with higher SOD activity and higher carotenoid content, since as lipid antioxidants, carotenoids protect membranes from oxidative damage [59]. Thus, our results point out that presence of GO seemed to alter shoot strategy to tolerate drought. In the absence of GO nanosheets and especially in the presence of bacteria (condition R), cells increased chlorophyll content and therefore photosynthetic activity must have increased. Photosynthesis is one of the cellular process that contributes most to ROS generation [60], explaining why, in a drought situation, plants not exposed to GO (Ctl and R conditions) showed higher oxidative stress and damage than those exposed to GO (GO and R + GO conditions). Photosynthesis also produces reduced organic compounds that can accumulate in the chloroplast (starch), leaving cells with more resources to induce metabolic pathways (proteins) to adapt cells to osmotic stress (higher proline content) and to control oxidative stress (increased carotenoid content and SOD activity), which, however, were not sufficient to suppress the effects of increased ROS and damage (increased LPO) that emerged. Exposure to GO decreased the content of chlorophylls and seemed to decrease photosynthetic activity (starch decrease) and oxidative stress, preventing membrane damage (lower LPO), despite the lower carotenoid content. Thus, addition of GO nanosheets and inoculation with bacteria appear to provide a similar and significant degree of protection against drought to shoots yet using different strategies that, when applied simultaneously, are not additive but cause lower disturbance in cell metabolism.

## 5. Conclusions

Although GO nanosheets evidenced an ability to disturb root metabolism and growth, in crops such as maize, which are grown for the production of biomass for forage or grain (aboveground organs), the effect of GO nanosheets was negligible under the present experimental conditions. However, the main effects of GO nanosheets were in a drought context, with the osmotic and antioxidant protection conferred by GO nanosheets to drought shoots, leading to higher shoot biomass. This is relevant from an agronomic and economic point of view, in terms of reducing economic losses in agricultural systems with lower water availability or in years with more severe droughts. PGPB inoculation in addition to GO nanosheets was shown to be relevant in drought plants, since it reduced the metabolic disturbance teased by GO nanosheets. The results of this study are thus promising and point to the possible use of GO nanosheets (alone or in combination with PGPB inoculation) as a sustainable methodology to reduce the impact of drought on important crops such as maize. These results are significant in the context of climate change, in which water becomes an increasingly scarce resource, especially in summer months, when maize is grown, rainfall is rare, and irrigation is imperative.

## Figures and Tables

**Figure 1 nanomaterials-11-00771-f001:**
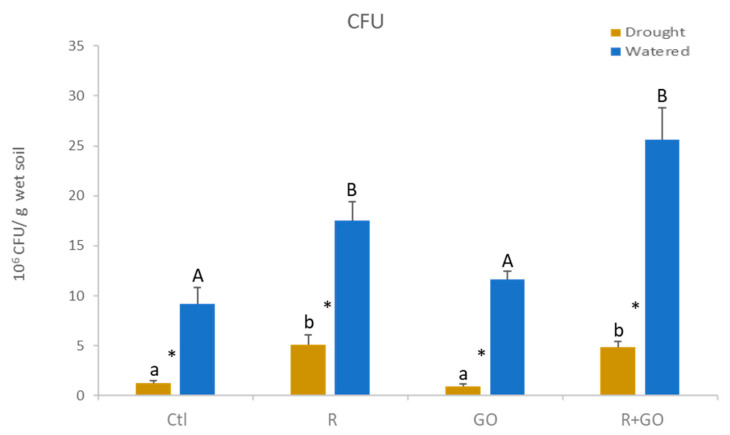
Colony forming units (CFU) in watered (blue bars) and drought (brown bars) sand where maize plants grew for 28 days at different conditions. Ctl—no addition of graphene oxide nanosheets, no bacterial inoculation; R—no addition of graphene oxide nanosheets, inoculation with *Rhizobium* strain E20-8; GO—addition of graphene oxide nanosheets, no bacterial inoculation; R + GO—addition of graphene oxide nanosheets and inoculation with *Rhizobium* strain E20-8. Values are means of three replicates + standard error. Different uppercase letters indicate significant differences (*p* < 0.05) among conditions in watered sand, different lowercase letters indicate significant differences among conditions under drought, and asterisks indicate significant differences between watered and drought for the same condition.

**Figure 2 nanomaterials-11-00771-f002:**
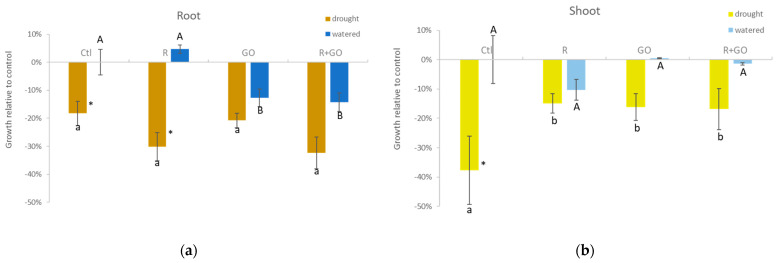
Root (**a**) and shoot (**b**) growth relative to control (condition Ctl) of maize plants watered (blue bars) or drought (brown bars) grown at different conditions. Ctl—no addition of graphene oxide nanosheets, no bacterial inoculation; R—no graphene oxide nanosheets addition, inoculation with *Rhizobium* strain E20-8; GO—addition of graphene oxide nanosheets, no bacterial inoculation; R + GO—addition of graphene oxide nanosheets and inoculation with *Rhizobium* strain E20-8. Values are means of at least six replicates ± standard error. Different uppercase letters indicate significant differences (*p* < 0.05) among conditions in watered plants, different lowercase letters indicate significant differences among conditions in drought plants, and asterisks indicate significant differences between drought and watered plants for the same condition.

**Figure 3 nanomaterials-11-00771-f003:**
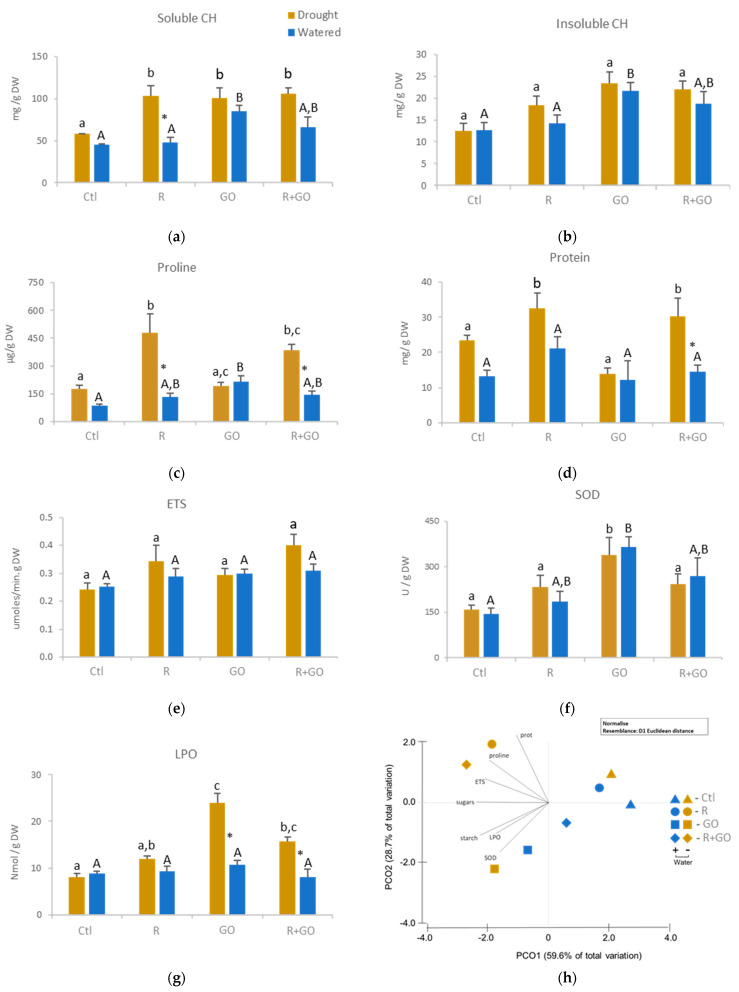
Biochemical parameters of the roots of watered (blue bars) or drought (brown bars) maize plants grown at different conditions (Ctl—no addition of graphene oxide nanosheets, no bacterial inoculation; R—no addition of graphene oxide nanosheets, inoculation with *Rhizobium* strain E20-8; GO—addition of graphene oxide nanosheets, no bacterial inoculation; R + GO—addition of graphene oxide nanosheets and inoculation with *Rhizobium* strain E20-8). (**a**) Soluble carbohydrates (CH); (**b**) insoluble carbohydrates; (**c**) proline; (**d**) protein; (**e**) electron transport system activity (ETS); (**f**) superoxide dismutase activity (SOD); (**g**) lipid peroxidation (LPO); (**h**) principal coordinates ordination of biochemical parameters in watered and drought plants at different conditions (Ctl, R, GO and R + GO). Values are means of three replicates + standard error. Different uppercase letters indicate significant differences (*p* < 0.05) among conditions in watered plants, different lowercase letters indicate significant differences among conditions in drought plants, and asterisks indicate significant differences between drought and watered plants for the same condition.

**Figure 4 nanomaterials-11-00771-f004:**
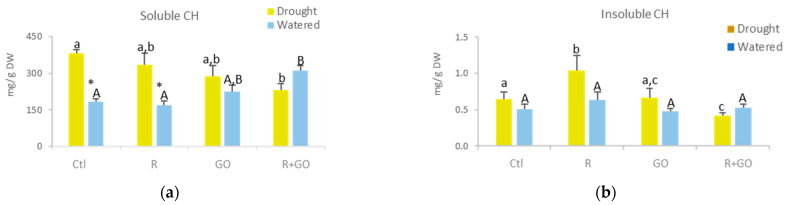
Biochemical parameters of the shoots of watered (light blue bars) or drought (light green bars) maize plants grown at different conditions (Ctl—no addition of graphene oxide nanosheets, no bacterial inoculation; R—no addition of graphene oxide nanosheets, inoculation with *Rhizobium* strain E20-8; GO—addition of graphene oxide nanosheets, no bacterial inoculation; R + GO—addition of graphene oxide nanosheets and inoculation with *Rhizobium* strain E20-8). (**a**) Soluble carbohydrates (CH); (**b**) insoluble carbohydrates; (**c**) proline; (**d**) chlorophyll a; (**e**) chlorophyll b; (**f**) carotenoids; (**g**) protein; (**h**) electron transport system (ETS) activity; (**i**) superoxide dismutase (SOD) activity; (**j**) lipid peroxidation (LPO); (**k**) principal coordinates ordination of biochemical parameters in watered and drought plants at different conditions (Ctl, R, GO and R + GO). Values are means of three replicates + standard error. Different uppercase letters indicate significant differences (*p* < 0.05) among conditions in watered plants, different lowercase letters indicate significant differences among conditions in drought plants, and asterisks indicate significant differences between drought and watered plants for the same condition.

## Data Availability

The data presented in this study are available on request from the corresponding author.

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
