# Peer review of "A Multifactorial Approach to Untangle Graphene Oxide (GO) Nanosheets Effects on Plants: Plant Growth-Promoting Bacteria Inoculation, Bacterial Survival, and Drought"

_nanomaterials, 2021, doi:10.3390/nano11030771_

Round 1

Reviewer 1 Report

The results are presented well. My major concerns are:

  1. TBARS assay: Even if the assay is widely used. The method in general is not precise. I recommed to use HPLC if possible.
  2. Frozen samples were milled in liquid nitrogen, followed by pestle and mortar homogenization in sodium phosphate buffer. Did the authors consider the use of protease inhibitors?

Author Response

Response to Reviewer

The authors would like to thank the reviewer for all comments and suggestions that greatly improved the quality and clarity of the manuscript. All comments were taken into consideration and the changes were done in the manuscript according to the reviewer suggestions. Changes are highlighted in yellow along the manuscript, except if they are too extensive, such as the Results section, that was completely rewritten.

Comment 1: TBARS assay: Even if the assay is widely used, the method in general is not precise. I recommend to use HPLC if possible.

Response to Comment 1: 

Thiobarbituric acid (TBA) not only reacts with malondialdehyde, a secondary product of lipid peroxidation, but also with fatty acid hydroperoxides, such as linoleic acid. Although the reaction with TBA does not discriminate between different lipid hydroperoxides that are formed under oxidizing conditions, it quantifies the total amount of hydroperoxides formed and  thus the level of damage in the membranes.

The HPLC technique allows identifying the different lipid hydroperoxides resulting from oxidative damage of membranes. However, in this study the total peroxidation level in each condition is enough to meet our objective, that was to evaluate the damage inflicted in membranes by each condition.

Comment 2: Frozen samples were milled in liquid nitrogen, followed by pestle and mortar homogenization in sodium phosphate buffer. Did the authors consider the use of protease inhibitors?

Response to Comment 2: The buffer used included EDTA, an inhibitor of metal proteases. In previous work the use of PMSF and E64 (two other common protease inhibitors for plant tissues) revealed to not have effect in protecting the activity of the enzymes assayed and interfered with other biochemical assays. Thus, only EDTA has remained in the buffer used, however even EDTA interferes with some biochemical assays (e.g. protein carbonylation), since in this study we did not evaluate this parameter, EDTA was included. The buffer used included PVP (polyvinylpyrrolidone) and DTT (dithiothreitol), which are reducing compounds that bind to phenolic compounds and avoid their oxidation. The oxidation of phenolic compounds (after tissue disruption in contact with O2 from air) inactivate enzymes. Since plant tissues may have high content of phenolic compounds the inclusion of PVP and DTT are essential for enzymatic assays in plant tissues.

Reviewer 2 Report

This study looks to link GO and PGPR effects on plant drought response by measuring some plant physical and biochemical parameters. In its present form I do not believe this manuscript is suitable for publication and it also requires significant revision of English language.

Specific comments:

Fig 2 is missing 2c and 2d

Why was this Rhizobium spp chosen as PGPR - is their previous data to support its efficacy as a PGPR?

Data from Fig 1is meaningless - this reflects total CFU and given the growth conditions are not sterile this will represent any contaminants etc as well. Could selective media for RNB be used here?

Please use scientific notation throughout (ie 2 x 10^6 cells, not 2 million cells)

Assessment levels of gene expression of key genes involved in abiotic stress response will be informative

Author Response

The authors would like to thank the reviewer for all comments and suggestions that greatly improved the quality and clarity of the manuscript. All comments were taken into consideration and the changes were done in the manuscript according to the reviewer suggestions. Changes are highlighted in yellow along the manuscript, except if they are too extensive, such as the Results section, that was completely rewritten.

Comment 1: Figure 2 is missing 2c and 2d.

Response to Comment 1: We didn’t understand why the reviewer refers the missing 2c and 2d in figure 2. This figure just presents the relative growth of the root (2a) and shoot (2b) and we did not find a reference to 2c and 2d.

Comment 2: Why was this Rhizobium spp chosen as a PGPR- is their previous data to support its efficacy as a PGPR?

Response to Comment 2: In previous works performed in our lab, we have data indicating this strain as a plant growth promoter. In the work presented by Cardoso (2019) (Cardoso, “Bacterial-induced plant growth promotion under drought: the importance of airborne communication”, Ph.D dissertation, Biology department, Univ. Aveiro, Portugal, 2019. Available: http://hdl.handle.net/10773/30179), it is reported the induction of plant growth by bacterial volatile organic compounds of this strain under different osmotic conditions.

Comment 3: Data from figure 1 is meaningless- this reflects total CFU and given the growth conditions are not sterile this will represent any contaminants etc as well. Could selective media for RNB be used here?

Response to Comment 3: In the environment no sterile conditions are present, thus we intentionally did not use sterile conditions. Also, there isn’t a selective medium only for RNB. The figure (figure 1) shows that the presence of GO did not reduce bacterial density in the substrate (inoculated or not) and that even could have a positive effect in the CFU numbers in watered substrate inoculated with Rhizobium (R+GO condition), and that drought had a high negative impact on bacterial survival. If the reviewer still finds the information meaningless, the figure could integrate the supplemental material.

Comment 4: Please use scientific notation throughout (ie 2 x 10^6, not 2 million cells).

Response to Comment 4:  The authors thank the reviewer suggestion. All the sentences where scientific notation was missing, were changed according to the suggestion proposed.

Comment 5: Assessment levels of gene expression of key genes involved in abiotic stress response will be informative.

Response to Comment 5: The authors found the awareness to assess the levels of expression of a gene set in response to abiotic stress an excellent idea for future work. In this study the parameters tested (7 for roots and 10 for shoots) give a good idea of the impact imposed by each condition to the physiology and biochemistry of maize plants, such as, production of photosynthates (chlorophylls, insoluble carbohydrates), osmotic adaptation (soluble carbohydrates, proline), metabolic activity (ETS), changes in metabolism (proteins), antioxidant response (carotenoids, SOD), cell damage (LPO).

Reviewer 3 Report

In this manuscript Lopes and the co-workers have presented their results on the study of individual and combined effects of graphene oxide (GO) and plant growth promoting bacteria Rhizobium sp. On different biochemical parameters of roots and shoots of exposed maize plant in normal and drought conditions. The study is very interesting and seems to be well designed and conducted. Obtained results are very intriguing. However, presentation and the description of the results should be modified and improved. Discussion should also be improved.

GENERAL REMARKS

The writing should be precise and accurate! Authors should be consistent in using terms and expressions.

English language should be improved. I advise using a help from the native speaker or professional English editing service

SPECIFIC REMARKS

Materials and methods

Some characteristics for the GO applied should be included in the manuscript, like particle size, if it is a monolayer or multilayer; layer thickness etc.

Materials and methods section does not seem to be fully correlated with the Results section. For example, according to the M&M author have measured the war activity, but didn’t’ present any results; they have also measured dry weight of whole, but one cannot find the results of these measurements; moreover, in the results section they have presented differences in root and shoots growth, but in the M&M failed to explain how these parameters was measured!

Results

Presentation of the Biochemical alterations in roots and shoots results is rather confusing; in Figures authors have chosen to describe the results grouped by parameters (each graph is presenting results of one parameter of all conditions); however, in the Results section they have chosen to group results of all parameters for one condition, which makes it difficult to follow. Therefore, the Results section needs to be rewritten. If, however, authors insist on the current grouping of the results in the text, then it would be easier to present all parameters in two tables, one for roots and one for shoots.

Authors should be more precise and accurate in writing.

Moreover, the whole results section is written in the form of Results and discussion; since this paper already contains Discussion, any discussion of the results and drawing any conclusion should be placed in the Discussion section rather than in Results.

Do authors have any indication if GO have entered root cells and if there is any difference between watered and drought-exposed plants? Have you performed any accumulation analyses? Or maybe tried to analyse root cells with electron microscopy in order to detect GO particles and/or to see changes in ultrastructure?

Discussion

Discussion is also a bit confusingly written. Authors should be more focused on their own results and try not to elaborate results of other studies in many details particularly if they have not measured the same parameters.

Considering the oxidative stress, it would be interesting to read why authors have chosen lipid peroxidation, and particularly SOD, since there are numerous antioxidant enzymes.

Figures

All Figures: x- and y-axis should both be visible on the graphs; it is rather inconvenient that authors have chosen the same mark (upper case letters) for different conditions tested in the experiment as for the statistics

Figure 1 – there is no need for two decimals in the y-axis values

Figure 2 – conditions marks (A to D) should not be above but below the graph since in the current version there is a possible confusion between marks for different conditions and marks for statistics

Marks of each graph presented on this figure should be placed in the left-up corner

Figure 3 – marks of each graph presented on this figure should be placed in the left-up corner;

On the figure b) statistic marks are missing

On the figure c) statistics seem strange; there is “ac” and “bc” but there is not “c”? It should be checked!

On the figure e) two decimals in the y-axis values should be omitted and the unit is not correct

Figure 4 – marks of each graph presented on this figure should be placed in the left-up corner;

On the figure b) the watered column of the condition D is missing

figures d) and e) I suggest to present them as total chlorophylls on one graph

figure h) check the unit; it is not OK!

Figure S1 – it is not clear what does the values on the y-axis represent? Are there any units?

All other specific remarks are included in the pdf of the manuscript.

Author Response

The authors would like to thank the reviewer for all comments and suggestions that greatly improved the quality and clarity of the manuscript. All comments were taken into consideration and the changes were done in the manuscript according to the reviewer suggestions. Changes are highlighted in yellow along the manuscript, except if they are too extensive, such as the Results section, that was completely rewritten.

Material and Methods

Comment 1: Some characteristics for the GO applied should be included in the manuscript, like particle size, if it’s a monolayer or a multilayer; layer thickness, etc;

Response to Comment 1: The following information was included in the text:

(Lines 105-109) “A commercial graphene oxide (GO) water dispersion (0.4 wt% concentration) was pur-chased from Graphenea, San Sebastian, Spain and used as received. According to our characterization by Atomic Force Microscopy (AFM, VEECO Multimode; USA), the GO nanosheets are mainly multilayer with an average thickness of 1.2 μm corresponding to 2 or 3 layers and a lateral size with a wide range of dimensions from 300 nm to 5 μm.”

Comment 2: Materials and methods section does not seem to be fully correlated with the results section. For example, according to the M&M author have measured the water activity, but didn’t present any results; they have also measured dry weight of whole, but cannot find the results of these measurements, moreover, in the results section they have presented differences in root and shoots growth, but in M&M failed to explain how these parameters was measured.

Response to Comment 2: The authors found the comment relevant. The figure containing the water activity was added as supplemental material and a new topic (lines 240-245) was included in the Results section (3.1-Water activity), describing the results obtained for water activity.

Additional description was included (lines 151-156) to clarify the procedure used to obtain root and shoot dry weight and to calculate relative growth.

Results

Comment 3: Presentation of the biochemical alterations in roots and shoots results is rather confusing: in figures authors have chosen to describe the results grouped by parameters (each graph is presenting results of one parameter of all conditions); however, in the results section they have chosen to group by results of all parameters for one condition, which makes it difficult to follow. Therefore, the Results section needs to be rewritten. If, however, authors insist on the current grouping of the results in the text, then it would be easier to present all parameters in two tables, one for roots and one for shoots.

Authors should be more precise and accurate in writing.

Moreover, the whole results section is written in the form of Results and discussion; since this paper already contains Discussion, any discussion of the results and drawing any conclusion should be placed in the Discussion section rather than in Results.

Response to Comment 3: The reviewer comments made us realize the difficulty to confirm the results in the figures while reading the Results section. Thus, the all Results section was rewritten to meet the reviewer suggestion. In Sections 3.4 (Biochemical alterations in roots, that corresponds to results in figure 3) and 3.5 (Biochemical alterations in shoots, that corresponds to results in figure 4) the subtitles are now the parameters analyzed and not the conditions tested.

Comment 4: Do authors have any indication if GO have entered root cells and if there is any difference between watered and drought-exposed plants? Have you performed any accumulation analyses? Or maybe tried to analyze root cells with electron microscopy in order to detect GO particles and/or to see changes in ultrastructure?

Response to Comment 4: This is a pertinent comment. We freeze dried roots in the two water regimes and exposed and non-exposed to GO and tried to identify GO nanosheets by SEM, but our attempt was not successful.

Discussion

Comment 5 (Comment in the PDF): This not relevant since you have not measured ABA; it is enough just to state that these authors also recorded osmotic stress in B. napus plants exposed to GO

Response to Comment 5: The sentence stating that ABA is increased by GO due to induction of osmotic stress was deleted. Since, as suggested by the reviewer, the direct induction of osmotic stress by GO was already pointed out by other studies (ref 45 and 48). The name of the species was also deleted.

Comment 6 (Comment in the PDF): English!

Response to Comment 6: Line 509– Corrected

Comment 7 (Comment in the PDF): How come drought is the second stress? What is the first stress? Confusing!

Response to Comment 7: Line 521 (former line 515) – The text was changed to meet reviewer suggestion to “to GO and drought”

Comment 8 (Comment in the PDF): Did Zhou and Hu found the upregulation of SOD or some other antioxidant enzyme(s)?

Response to Comment 8: Line 529 – Zhou and Hu (ref 55) also determined the activity of peroxidase, thus this enzyme was also included in the text.

Comment 9 (Comment in the PDF): Did they showed images of damaged cells?

Response to Comment 9: Line 530 – The damage is not an injury but rather oxidative damage. The sentence was rewritten to avoid misinterpretation “GO induced oxidative stress that caused damage in plant cells.”

Comment 10 (Comment in the PDF): Unclear. Does this sentence refer to the results obtained with GO only? Because on this treatment (Condition C) lipid peroxidation was not significantly different compared to GO+bacteria

Response to Comment 10: Lines 531-534–   As the reviewer indicates these results are from the condition GO (formerly designated as condition C) and the sentence is comparing the results obtained in the presence and absence of drought. In order to make the sentence clearer it has been rewritten. "These results are in agreement with the oxidative damage induced by GO in the roots of Zea mays observed in our study. In the presence of GO and drought root cells were unable to further increase SOD activity and to manage a higher level of stress, as LPO increase evidences.”

Comment 11 (Comment in the PDF): I dont't see the relevance since in your study you have not measured AB. It could be mentioned but in a much shorter version

Response to Comment 11: Lines 545-551– The text was shortened in order to meet reviewer suggestion. The text included is: “The decrease in chlorophylls and carotenoids originated by GO, observed in our study, may be related to resources diversion for the synthesis of other metabolically related compounds such as ABA, that has already been reported to increase in plants exposed to GO [16]. Since ABA induces stomata closure [49], reducing gas exchange and CO2 concentration in the mesophyll, the Calvin cycle activity and photosynthates production will be lower [49], leading to a decrease in storage energy (insoluble carbohydrates).”

Comment 12 (Comment in the PDF): But you didn't performed any microscopy study to be able to state that you have detected cell damage

Response to Comment 12: Line 553– Once more we were talking about oxidative damage. The type of damage (oxidative damage) was identified in the text in order to avoid misinterpretation.

Comment 13 (Comment in the PDF):  compared to what?

Response to Comment 13: Lines 560-562– In order to clarify the sentence, it was changed to “explaining why in a drought situation plants not exposed to GO (Ctl and R conditions) showed higher oxidative stress and damage than those exposed to GO (GO and R+GO conditions).

Comment 14 (Comment in the PDF): But you didn't measure ABA! you didn't measure ROPS generation

Response to Comment 14: Lines 567-568 – The text was changed to meet reviewer comments to “Exposure to GO decreased the content of chlorophylls and seemed to decrease photosynthetic activity (starch decrease) and oxidative stress,”

General comments on Discussion Section

Comment 15: Discussion is also a bit confusingly written. Authors should be more focused on their own results and try not to elaborate results of other studies in many details particularly if they have not measured the same parameters.

Response to Comment 15: The answer and changes to comments made in the PDF to the Discussion section responded to more specific comments, but ended up eliminating many of the statements that included unmeasured parameters and resulted in a discussion more focused.

Comment 16: Considering the oxidative stress, it would be interesting to read why authors have chosen lipid peroxidation, and particularly SOD, since there are numerous antioxidant enzymes.

Response to Comment 16: SOD is the first antioxidant response to oxidative stress and even at mild oxidative stress alterations in SOD activity can be detected. On the other hand, GO nanosheets were reported to damage membranes and to induce oxidative stress. Thus LPO, a parameter that measures oxidative membrane damage, seemed to be a pertinent parameter to determine. 

Figures

Comment 17: All Figures: x- and y-axis should both be visible on the graphs; it is rather inconvenient that authors have chosen the same mark (upper case letters) for different conditions tested in the experiment as for the statistics

Response to Comment 17: Figures were changed to meet the reviewer suggestion. All graphs have visible axis. Also the condition name was changed to avoid confusion with statistical significance indication (upper case letters). Thus, condition A, B C and D were changed to Ctl, R, GO and R+GO, respectively both in figures and in the text

Comment 18: Figure 1 – there is no need for two decimals in the y-axis values.

Response to Comment 18: Changed according to reviewer suggestion

Comment 19: Figure 2 – conditions marks (A to D) should not be above but below the graph since in the current version there is a possible confusion between marks for different conditions and marks for statistics

Response to Comment 19: The denomination of conditions was changed to avoid confusion with statistical significance indication (upper case letters).

Comment 20: Marks of each graph presented on this figure should be placed in the left-up corner

Response to Comment 20: The position of the marks of each graph is a rule of the journal rather than a choice of the authors

Comment 21: Figure 3 – marks of each graph presented on this figure should be placed in the left-up corner

Response to Comment 21: Already answered in Comment 20.

Comment 22: On the figure 3b) statistic marks are missing

Response to Comment 22: The statistics of figure 3b is now included

Comment 23: On the figure 3c) statistics seem strange; there is “ac” and “bc” but there is not “c”? It should be checked!

Response to Comment 23: The statistical analysis showed that the proline levels for R+GO are significantly different from Ctl and those of GO are significantly different from R, but the levels at GO are not significantly different from R+GO, and therefore have to be identified with the same letter

Comment 24: On the figure e) two decimals in the y-axis values should be omitted and the unit is not correct

Response to Comment 24: The decimals and units of figure 3e are now correct

Comment 25: Figure 4 – marks of each graph presented on this figure should be placed in the left-up corner.

Response to Comment 25: As mentioned earlier the position of the marks of each graph is a rule of the journal rather than a choice of the authors

Comment 26: On the figure b) the watered column of the condition D is missing

Response to Comment 26: Figure 4d is now corrected and the column corresponding to condition R+GO (former condition D) is now visible

Comment 27: figures d) and e) I suggest to present them as total chlorophylls on one graph

Response to Comment 27: We understand the reviewer's suggestion, since the variation among conditions is similar. However, there is a difference that in our opinion hinders the joining of the two graphs. In chlorophyll a, there are no significant differences among any of the conditions tested, while in chlorophyll b there are significant differences among some conditions in plants exposed to drought.

Comment 28: figure h) check the unit; it is not OK!

Response to Comment 28: The units of figure 3h are now corrected

Comment 29: Figure S1 – it is not clear what does the values on the y-axis represent? Are there any units?

Response to Comment 29: Water activity has no units as it is a relative value to “pure” water, arbitrarily assigned the value 1.

Round 2

Reviewer 1 Report

The authors have addressed my points.